# p53 and Its Isoforms in Renal Cell Carcinoma—Do They Matter?

**DOI:** 10.3390/biomedicines10061330

**Published:** 2022-06-06

**Authors:** Agata Swiatkowska

**Affiliations:** Institute of Bioorganic Chemistry, Polish Academy of Sciences, 61-704 Poznan, Poland; agaswiat@ibch.poznan.pl

**Keywords:** p53 protein, p53 isoforms, von Hippel-Lindau protein (VHL), hypoxia-inducible factor (HIF), renal cell carcinoma (RCC)

## Abstract

p53 is a transcription al factor responsible for the maintenance of cellular homeostasis. It has been shown that more than 50% of tumors are connected with mutations in the *Tp53* gene. These mutations cause a disturbance in cellular response to stress, and eventually, cancer development. Apart from the full-length p53, at least twelve isoforms of p53 have been characterized. They are able to modulate p53 activity under stress conditions. In 2020, almost a half of million people around the world were diagnosed with renal cancer. One genetic disturbance which is linked to the most common type of kidney cancer, renal cell carcinoma, RCC, occurs from mutations in the *VHL* gene. Recent data has revealed that the VHL protein is needed to fully activate p53. Disturbance of the interplay between p53 and VHL seems to explain the lack of efficient response to chemotherapy in RCC. Moreover, it has been observed that changes in the expression of p53 isoforms are associated with different stages of RCC and overall survival. Thus, herein, an attempt was made to answer the question whether p53 and its isoforms are important factors in the development of RCC on the one hand, and in positive response to anti-RCC therapy on the other hand.

## 1. Introduction

Kidney cancer (also called renal cancer) is one of the most common tumors, aside from bladder cancer of the urinary system [1]. In 2020, there were 431,288 new cases, and mortality statistics showed 179,368 death cases occurred worldwide, according to the Global Cancer Observatory of the International Agency for Research of Cancer, IARC (https://gco.iarc.fr/, accessed on 1 May 2022). Consequently, kidney cancer is the 16th most common cancer globally (Figure 1A). The statistics are even worse in the case of men (Figure 1B). In 2020, there were 271,249 new diagnosed cases with 115,600 new death cases. Asia and Europe are two regions with the highest rate of incidence, with 156,470 and 138,611 reported cases, respectively, in 2020 (https://gco.iarc.fr/, accessed on 1 May 2022). 

There are several types of kidney cancer. The most common type, however, is renal cell carcinoma (RCC) which arises from renal tubular epithelial cells, and it accounts for approximately 90–95% of all kidney cancer cases [1,2]. The main subtypes of renal cell carcinoma are clear cell (ccRCC), papillary (pRCC), and chromophobe (chRCC) renal cell carcinomas [1,2]. Renal medullary carcinoma and sarcomatoid carcinoma occur less frequently [2]. Another type of kidney cancer is called Wilms’ tumor and it affects children [3]. In spite of the improvement of diagnostic methods, preventive and screening tests recent years have shown a steady rise in kidney cancer incidence [4].

It has been observed that one of genetic factors leading to RCC are mutations in the von Hippel–Lindau (VHL) tumor suppressor gene placed in chromosome 3p [5,6]. Alterations in the *VHL* gene result in the deregulation of expression of the hypoxia-inducible factor (HIF), which plays a crucial role as a transcription factor in oxygen regulation [7]. As a consequence of an increase in the HIF protein level, the expression of genes involved in the hypoxia pathway is changed [7,8]. Eventually, activation of the genes engaged in the hypoxia response, angiogenesis, and other genes including the vascular endothelial growth factor (VEGF), is observed [7]. 

However, it is known that p53 protein is a major transcriptional factor involved in the general stress response pathway and tumorigenesis prevention [9]. In response to different stress conditions, such DNA damage, hypoxia, or radiation, p53 induces genes to restore cell homeostasis or apoptosis, which leads to the arrest of the cell cycle progression [10,11]. Mutations in the *Tp53* gene result in p53 dysfunction and disruption of the p53-protein network in the cell [12]. Particularly hotspot mutations, which occur with a high frequency in the *Tp53* gene, contribute to the cancerous phenotype [12]. These mutations, mostly located in the DNA-binding domain of p53, result in the production of a faulty protein which loses its characteristics such as the ability to properly interact with DNA and protein partners. On the other hand, it has been demonstrated that mutant p53 is capable of gaining new characteristics to promote an altered transcriptional pattern and metabolism, cellular invasion, and migration [13,14]. 

Does mutant p53 contribute to renal cancer development or do mutations in this gene occur sporadically in kidney cells? Since the expression of the *Tp53* gene results in at least twelve p53 isoforms, a question arises concerning their involvement in renal cancer growth and spread. In the present article, the latest knowledge concerning the role of p53 and its mutated version, as well potential functions of p53 isoforms in terms of the development and progression of kidney cancer, are discussed. Special attention is focused on renal cell carcinoma as the most common kidney cancer. 

## 2. Wild-Type and Mutant p53 in Renal Cell Carcinoma

### 2.1. Wild Type p53 and Von Hippel–Lindau—Alone and Together

The p53 tumor suppressor protein is one of “gatekeepers” in the cell which plays an essential role in stress response [9,15]. In a healthy cell, the p53 level is low due to its ubiquitination by MDM2, an E3 ubiquitin ligase, and subsequent degradation. In response to different stress conditions p53 is phosphorylated in order to prevent the Mdm2-p53 interaction, and consequently, the p53 level increases [9]. The transcriptional activity of p53 induces genes which result in the cell cycle progression arrest or apoptosis [10]. Proper functioning of p53 is the basis of effective chemotherapy and radiotherapy in anti-cancer treatments [9]. The von Hippel–Lindau protein is also a tumor suppressor which is able to interact with many proteins, which results in its engagement in cellular processes, including angiogenesis and cell metabolism [16]. The VHL protein is also involved in the cell response to oxygen via its binding to the hypoxia-induced factor, HIF, which leads to the ubiquitination and subsequent degradation of HIF [16]. Mutations of the *VHL* gene cause alteration in HIF expression, and as a consequence, HIF activates its target genes responsible for energy metabolism and angiogenesis. An increased activity of HIF results in the formation of new blood vessels around the tumor to promote its progression [17,18].

It has been observed that in numerous cases of RCC, in which the *VHL* gene is mutated, the expression of proteins linked to the p53-response pathway is deregulated, or p53 is non-functional despite its wild-type status (WTp53) [19,20]. A question has arisen as to whether VHL and p53 could somehow be dependent on each other when performing their roles in the cell. Roe and colleagues have proposed a model which has at least partly answered that question [21,22]. VHL protein possesses two domains: (i) α domain which interacts with elongin C, and (ii) β domain which binds the hypoxia-inducible factor, HIF [23]. It has been found that the VHL protein is able to interact with p53 protein via its α domain (Figure 2) [21]. This VHL-p53 interaction stabilizes p53 and prevents it from MDM2-mediated ubiquitination, as well as blocking the nuclear export of p53 [21]. It is likely that VHL bound to ATM kinase facilitates phosphorylation of p53, and as a consequence, indirectly inhibits p53 degradation mediated by MDM2 (Figure 2). Moreover, VHL enhances acetylation of p53 protein. In VHL-deficient cells, p53 stabilization is reduced upon genotoxic stress. This indicates that VHL is a crucial factor in the VHL-ATM-p53 complex formation under stress conditions (Figure 2) [21]. Importantly, it has been demonstrated that the VHL and p53 interplay results in activation of p53 downstream genes such as *p21* and *Bax* upon DNA damage [21]. Such enhancement of p53 transcriptional activity by VHL eventually leads to cell cycle arrest and apoptosis in the A498 cell line, which is stably expressed in the wild-type VHL (A498/VHL) [21]. 

The model provided by Roe and colleagues indicated a new function of the VHL protein, with regard to the p53-depenedent response to stress factors in kidney cells. It could also explain why VHL-deficient cells of RCC are resistant to chemotherapy even if they exhibit wild-type p53; however, it also brought new questions regarding a potential, more general role of VHL in p53-mediated stress response, particularly, in the case of other types of cancers. More recently, it has also been shown that VHL and p53 act in a synergic way to increase the response to chemotherapy in clear cell renal cell carcinoma [24]. Co-expression of VHL and p53 has been noted to lead to a higher rate of G0/G1 cell arrest and apoptosis in cell lines from ccRCC in ADM (Adriamycin) or sunitinib treatment [24]. The authors even suggest that VHL and p53 could be molecular markers to identify ccRCC patients who would benefit from chemotherapy based on ADM or sunitinib [24].

Diesing and colleagues have noticed that the levels of total WTp53 and its phosphorylated form are elevated by irradiation; however, no increased transcriptional activity of p53 or changes in cell proliferation and migration have been observed in applied ccRCC cell lines [25]. The authors have explained these observations, similarly to Roe and colleagues, by the dysfunction of genes which protein products are involved in for the activation pathway of p53, including, *VHL* or *PBRM1* [25,26]; however, the authors have also suggested that inhibition of p53 functionality might result from an impact of hypoxia on p53 protein. Hypoxia conditions lead to the accumulation of p53, and consequently, they may cause competition in gene activation between p53 and HIF factor [25].

Recently, an interesting model has been proposed in which enhancer RNA, RNA-p53-bound enhancer region 2 (p53BER2) is involved in the regulation of p53 activity in RCC [27]. Enhancer RNAs (eRNAs) are a group of non-coding RNAs which are produced by RNA polymerase II from certain enhancer sites, and they seem to be engaged in many cellular processes as well in tumorigenesis [28]. It has been demonstrated that p53-bound enhancer regions (p53BERs) produce eRNAs which are functional, since they are involved in an efficient transcriptional activity of p53 [29]. Xie and colleagues have shown that the level of p53BER2 RNA is much lower in renal cancer tissues than in normal kidney tissues [27]. Furthermore, the authors have implied that p53BER2 RNA could be involved in the p53-mediated cell cycle and senescence of renal cancer cells [27]. Since RCC exhibits an extremely low level of p53BER2 RNA, this might explain why WTp53 is functionally inactive. It has been suggested that an increase in p53BER2 RNA could reactivate the p53-p53BER2 RNA pathway in RCC; however, more research is needed to better characterize p53BER2 RNA with regard to renal cancer [27].

It is clear that p53, together with VHL, are important factors in terms of RCC development, but different observations show that the mechanism seems to be much more complicated, and other players including various proteins and non-coding RNAs are involved. 

### 2.2. Mutant p53 and Clinical Prognosis for RCC Patients

There is no doubt that mutations in the *Tp53* gene are linked to cancer development [12]. Interestingly, despite the fact that missense mutations of thr *Tp53* gene (a mutation in DNA which results in the wrong amino acid being incorporated into a protein) occur in 190 different codons, eight mutations account for approximately 28% of the total p53 mutations [12]. It has been observed that the most mutated region of *Tp53* gene comprises the DNA-binding domain. These eight mutations (hotspot mutations) occur in six positions: R175H, Y202C, G245S, R248Q, R248W, R273C, R273H, and R282W (Figure 3A) [12]. Mutations in the DNA-binding domain affect the basic function of p53, namely, the ability to interact with the promoter region of downstream genes (loss-of-function). This results in promoting cancer development by depriving cells of tumor suppressive responses, including senescence and apoptosis. On the other hand, p53 mutated cells gain selective new features as a consequence of the different activities of mutant p53, Mutp53, (gain-of-function). It appears that mutant p53 is able to reprogram cells so that they can cope with challenging conditions and promote survival, migration, metastasis, and chemoresistance [14]. It has been shown that mutant p53 is able to sustain glucose intake by the ability to induce membrane translocation of the glucose transporter GLUT1 [30]. Mutant p53 is also able to promote a pro-survival oxidative stress response that allows cancer cells to cope with high levels of intracellular reactive oxygen species (ROS) [31]. Proteome-wide analysis has revealed that Mutp53 promotes chromatin recruitment of PARP (Poly (ADP-ribose) polymerase), along with other DNA replication factors, to increase DNA replication efficiency [32]. It emerges that mutant p53 is able to reshape the entire transcriptome and proteome to promote cancer development and progression. Mantovani and colleagues described the mutant p53 as a guardian of the cancer cell [14] and they were correct to name it as such. 

How often mutations are in the *Tp53* gene/p53 protein (*Tp53*/p53) observed in renal cancer? Surprisingly, the level of mutations in *Tp53*/p53 seems to be quite low in kidney cancer compared to other cancers. According to the Catalogue Of Somatic Mutations In Cancer, COSMIC, those mutations are observed in 8.75% of total mutated samples from kidney (Table 1); however, Zhang and colleagues have shown that 69 (25.9%) out of 226 samples from renal cancer patients exhibit p53 mutations [33]. In earlier studies it has also been demonstrated that 49 (28%) out of 175 analyzed samples harbored p53 mutations [34]. This is probably due to the fact that the rate of mutations in *Tp53*/p53 in kidney cancer depends on the cohort used in different studies.

Distribution of mutations across histological subtypes of renal cell carcinoma, as a predominant form of kidney cancer, is varied, as shown in Table 1 and Figure 3. The highest rate of mutations at 25.62% is observed in chromophobe renal cell carcinoma (chRCC) (Table 1). Whereas in the case of clear cell renal carcinoma (ccRCC) and papillary renal cell carcinoma (pRCC), the mutation rate is at 6.73% and 1.98%, respectively (Table 1). Slightly different data have recently been presented by Li and colleagues; however, their statistics of *Tp53*/p53 mutations rate were based on the Broad GDAC Firehose database [4]. In the case of both applied databases, the highest frequency of mutations in *Tp53* gene occurs in chromophobe renal cell cancer (Table 1 and [4]). 

In order to look closer at the distribution of mutations in *Tp53*/p53 in renal cancer, and in particular, its histological subtypes, data from the COSMIC database was applied (Figure 3B–E). The most mutated region of *Tp53*/p53 in kidney cancer comprises the DNA-binding domain with hot spot missense mutations in arginine positions: R175, R248, and R273 (Figure 3A,B); however, mutations occurring in the oligomerization domain (OD) are also observed (Figure 3B). This domain is responsible for the formation of the transcriptionally active p53 tetramer [35]. It has been demonstrated that mutations in the oligomerization domain affect the tetramerization process of p53, and as a result, the pattern of expression of p53-depenedent genes is changed [36,37]. Mutations in the oligomerization domain of p53 are very often observed in Li–Fraumeni syndrome (LPS) which is a complex hereditary predisposition to the development of a wide range of cancer types [38]. In a majority of LFS patients, soft-tissue cancers, bone sarcomas, breast cancer, brain tumors, adrenocortical carcinoma, and acute leukemia, are observed [38]. Development of cancers in other organs, including kidney cancer, is also observed. Interestingly, two characteristic mutations in the p53 oligomerization domain at position R337 and R342 are observed in renal cancer (Figure 3B). Mutation at position R337 is observed mostly in chromophobe renal cell carcinoma (Figure 2C). Mutation of arginine at position 342 does not occur in renal cell carcinoma, but it almost exclusively appears in Wilms’ tumor which affects children [3]. There is no data as to whether kidney cancer development in patients bearing mutations in the oligomerization domain of p53 resulted from LPS, or whether they were not inherited and appeared de novo. The mutation patterns are different for different histopathological subtypes of renal cell carcinoma. In the case of chRCC, the highest rate of mutations is observed at positions R213; however, aside from missense mutations, frameshifts and nonsense mutations are detected at this position (Figure 3C). On the other hand, the p53 mutation pattern for ccRCC revealed that glycine 244, arginine 273 and proline 278 mutate with the highest frequency (Figure 3D). Missense mutations at positions K132 and C135 in clear cell renal cell carcinoma are also observed (Figure 3D). There is little data concerning papillary renal cell carcinoma, as only 8 samples are described in the COMIC database; however, mutations occur in different positions compared with chRCC or ccRCC (Figure 3E). The analysis revealed that each histopathological subtype of renal cell carcinoma is characterized by a specific p53 mutation profile or at least distinct positions in *Tp53*/p53 which are mutated at a higher rate (Figure 3B–E); however, there is a small amount of research focusing on the analysis of the p53 status in renal cell carcinoma patients, and therefore, there is not enough data that would give a complete picture of the mutation pattern of *Tp53*/p53 in this particular cancer. 

Moreover, how do mutations in p53 influence the development and progression of renal cell carcinoma? Earlier studies of Uhlman and colleagues have shown that positive p53 immunostaining in RCC is correlated with poor survival in patients with early-stage cancer [34]. Association of overexpression of p53 in samples from patients with a worse clinical prognosis and poorer overall survival has also been demonstrated by other research groups [8,39,40,41]. Since wild-type p53, WTp53, is kept at a low level in health cells under normal conditions [9], in most cases, it causes an overexpression of p53 which is visualized by positive immunostaining corresponds to mutation(s) in *Tp53*/p53 [4]; however, it has also been suggested that the observed overexpression of p53 in RCC concerns the wild-type, and an increased level of WTp53 protein is associated with reduced overall survival [42]. As has been discussed by Li and colleagues [4] in numerous studies, p53 antibodies have been used which do not distinguish between the wild-type and mutants of p53, that could make interpretations of the results more difficult. Nevertheless, an increase in the expression of wild type p53 in RCC has also been demonstrated [25] which we discussed in paragraph above.

Recently, Reig Torras and colleagues have shown that RCC patients harboring *Tp53*/p53 mutations, together with a mutation in the *SMARCA4* gene, exhibit a more aggressive tumor phenotype [43]. SMARCA4 (BRG1) protein is a part of the SWI/SNF complex, and it acts as a transitional activator [44]. It has been shown that BRG1 is able to interact with the p53 protein and is needed for its activity [44]. Patients with mutations in both genes, *Tp53* and *SMARCA4*, are poor candidates for active surveillance, which involves closely watching a patient’s condition but not giving any treatment or taking any action unless there are changes in the patient’s test results that show the condition is getting worse. The data presented by Reig Torras and colleagues [43] indicate that analysis of mutations in both the *Tp53* and *SMARCA4* genes could help select RCC patients for active surveillance, which gives patients more comfort in their lives by avoiding or at least delaying the need for chemotherapy or surgery.

Mutations in *Tp53*/p53 with regard to renal cell carcinoma seem to be detected less frequently than in other cancers, but on the other hand, there is relatively little data concerning the p53 status in this particular cancer; however, presence of the *Tp53*/p53 mutant has an impact on the development of RCC and on clinical prognosis. A comprehensive analysis of *Tp53*/p53 mutations in samples from different RCC patient cohorts is needed to better understand the association of the p53 status with clinical parameters, which, in the future would help improve treatment strategies for patients. 

## 3. p53 Isoforms in RCC

As well as the full-length p53, there are several isoforms of this protein which lack the N- or/and C-terminus [45] (Figure 4). They seem to play distinct functions and they can influence p53 to modulate the final outcome [46,47]. P53β and p53γ isoforms which result from alternative splicing events and possess shorter C-ends (Figure 4), have been shown to promote apoptosis in the breast cancer cell line, MCF-7 [48]. Interestingly, p53β isoform enhances full-length p53 activity on *p21* and *Bax* promoters, whereas p53γ is only able to stimulate Bax expression [48]. Moreover, in the presence of TG3003, an inhibitor of CDC2-like kinase 1 acting as a splice-modifying compound, both isoforms, p53β and p53γ, are able to maintain cell growth in a p53-dependent manner. This observation indicates dual functions of p53β and p53γ in the regulation of cell response [48]. Meta-analysis of 6 breast cancer cell lines, 148 breast cancers specimens, and 31 matched normal adjacent tissues revealed that p53β expression is negatively associated with tumor size and positively associated with disease-free survival [49]. Association of a high level of p53β with better prognosis have also been observed for acute myeloid leukemia patients [49,50]. Moreover, it has also been shown that p53β isoform plays a protective role in the case of breast cancer patients bearing a mutation in p53 [49].

Δ40p53 isoform lacks the first 39 amino acids from the N-end corresponding to the domain with the Mdm2 binding site, which results in an Mdm2-indepenedent stress response (Figure 4) [51,52,53]. Recently, it has been shown that Δ40p53 is able to trigger or inhibit p53 transcriptional activity depending on the cellular context [54]. Moreover, Δ40p53 induces many apoptosis-related genes which are not induced by p53 [55]. Overexpression of Δ40p53 has been observed in several tumorous tissues, including glioblastoma, melanoma, breast, and ovarian cancer [52]. Upregulation of Δ40p53 is likely to affect the p53 function, which results in the disturbance of the balance between growth control and pro-survival. 

Recently, more attention has been paid to the Δ133p53 isoform whose transcript starts from P2 promoter located within intron 4 of the human *Tp53* gene [56,57]. Since the Δ133p53 protein is synthetized from 133 methionine, it lacks the domain with the Mdm2 binding site and, partly, the DNA-binding domain (Figure 4). It remains unclear whether it is capable of directly binding to DNA and acting as a transcriptional factor. Interestingly, it has recently been shown that under normal conditions, Δ133p53 represses the transcription of p53 target genes associated with senescence but not the expression of apoptosis genes [58]. However, upon oxidative stress, Δ133p53 coordinates the activity of p53 to promote cell survival [59]. It has also been demonstrated that the Δ133p53 protein is engaged in carcinogenesis, including angiogenesis and metastasis, as well as in cellular events, including proliferation, cellular senescence, and apoptosis [56,60,61]. The Δ160p53 isoform’s second protein product, which is synthesized from a P2-initiated transcript, has been detected in several cell lines endogenously expressing different mutant p53s [62,63]. Moreover, wild-type p53 cell lines (HCT116, U2OS, and A549) show either no signs or low levels of Δ160p53 expression [64]. It seems that Δ160p53 is expressed mostly in mutant p53 cells promoting tumorigenesis [64]. It has also been shown that higher levels of Δ160p53 expression in melanoma patients are associated with cancer aggressiveness [63]. 

Are p53 isoform expression patterns associated with renal cancer development? Earlier studies using tissue samples of 45 RCC patients at different tumor stages and 25 non-neoplastic tissues have shown that expression of almost all p53 isoforms changed during cancer development and progression [65]. An increase in the p53β and p53γ mRNA levels have been observed in early stages of carcinogenesis (pT1 and pT2), whereas in pT3 carcinomas, the mRNA levels of these isoforms have been comparable to non-neoplastic tissues [65]; however, higher levels of Δ40p53 and Δ40p53γ have been detected in later tumor stages [65]. Upregulation of Δ40p53 expression has also been observed in RCC patients with mutant full-length p53 compared with those with wild-type p53; however, no difference has been found between normal and neoplastic samples [66]. Moreover, no association between the level of Δ40p53 and patient survival has been observed [66]. It has also been shown that cell lines derived from pT3 tumors, with or without mutation in the *Tp53* gene, exhibit different expression patterns of p53 isoforms, and their regulation with Topotecan (topoisomerase 1 inhibitor) treatment has been varied [65]. Thus, the authors have concluded that based on changes in the expression profiles of p53 isoforms, the p53 response to treatment in RCC could not be predicted [65].

More recently, it has been shown that p53β is associated with better recurrence-free survival, RFS (the length of time after primary treatment for a cancer ends that the patient survives without any signs or symptoms of that cancer), and overall survival [33]. Analysis of mRNAs from patients have revealed that a higher level of the expression of the p53β isoform is observed at a lower stage of tumor, pT1 [33]. This is in line with earlier observations showing an increase in both p53β and p53γ isoforms at pT1 and pT2 stages [65]. Interestingly, patients harboring a p53 mutation with a high level of p53β exhibit improved RFS (median 58.2 versus median 46.0) and overall survival compared with those who have a low level of p53β isoforms [33]. Further analysis of mRNAs from RCC patients has shown an increased level of Bax and caspase-3 in the case of patients with a high expression of p53β. Additionally, overexpression of the p53β isoform in 786-O and CAKi-1 cells has been shown to lead to a higher level of apoptosis regardless of p53 status [33]; however, Diesing and colleagues have shown no association between p53 isoforms, advanced tumor stages, and clinical features of RCC patients (a 55 patient cohort) [25]. They have only observed that a smaller tumor size was characterized by Δ133p53α (Δ133p53) expression [25]. It was an unexpected observation since Δ133p53 promotes cell survival and metastasis [56,60,61]. Discrepancies between the data could have at least, in part, resulted from different characteristics of patient cohorts taking part in different research projects.

It emerged that changes in p53 isoforms are somehow associated with different stages of RCC; in particular, a high level of p53β seems to be an important molecular indicator of better prognosis for patients. Analyses of p53 isoform patterns in RCC patients could help predict how cancer will progress and what treatment is potentially effective.

## 4. Conclusions

It has been shown that the crosstalk between p53 and VHL is a pivotal element in the DNA-damage response mediated by p53 in RCC [21]. VHL-deletion seems to be responsible, at least partly, for a decrease in p53 activity, which is often observed in RCC [21]. Moreover, potential involvement of eRNA (p53BER2 RNA) in a p53-dependent pathway adds an additional new layer of p53 regulation in renal cancer and indicates new directions of research [27]. In spite of the fact that the *Tp53* gene is infrequently mutated in RCC, the presence of a mutated version of p53 influences clinical prognosis and overall survival [4,39,40,41]. Immunotherapies which target p53 mutants could be a new direction in RCC treatment [67]. It seems that p53 isoforms also have an impact on clinical parameters. Of particular interest are observations concerning the p53β isoform, indicating that its high expression is positively associated with an elevated level of Bax and caspase-3 and a higher rate of apoptosis of RCC cells [33,65]. The p53β isoform level could be considered as a marker which might indicate a better chance for overall survival and RFS. Additionally, the controlled modulation of the expression pattern of p53 isoforms, by antisense oligonucleotides at the RNA level [53,57], and by small molecules at the protein level [68], which could be an alternative approach in anti-cancer therapy applied in RCC in the future. Moreover, monitoring of changes in expression profiles of the specific isoforms including p53β, p53γ, Δ133p53, and Δ160p53 might be applied in a personalized medicine to increase therapy effectiveness [33,63,65]. Altogether, p53 and its isoforms seem to mediate cell response and cell sensitivity to treatment in renal cell carcinoma; however, more research is needed to better understand the role of network interactions between p53 and its isoforms with other proteins and RNAs in this cancer.

## Figures and Tables

**Figure 1 biomedicines-10-01330-f001:**
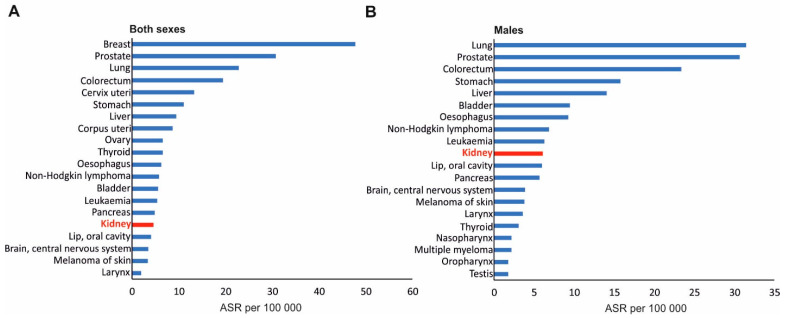
Twenty types of cancer with the highest incidence rates in 2020, worldwide, in both sexes (**A**) and in men (**B**). ASR (age-standardized rate) is a weighted mean of the age-specific rates where the weights are taken from the population distribution of a standard population; the ASR is expressed per 100,000. Kidney cancer is marked in red. Based on data from the Global Cancer Observatory of the International Agency Research of Cancer, IARC (https://gco.iarc.fr/, accessed on 1 May 2022).

**Figure 2 biomedicines-10-01330-f002:**
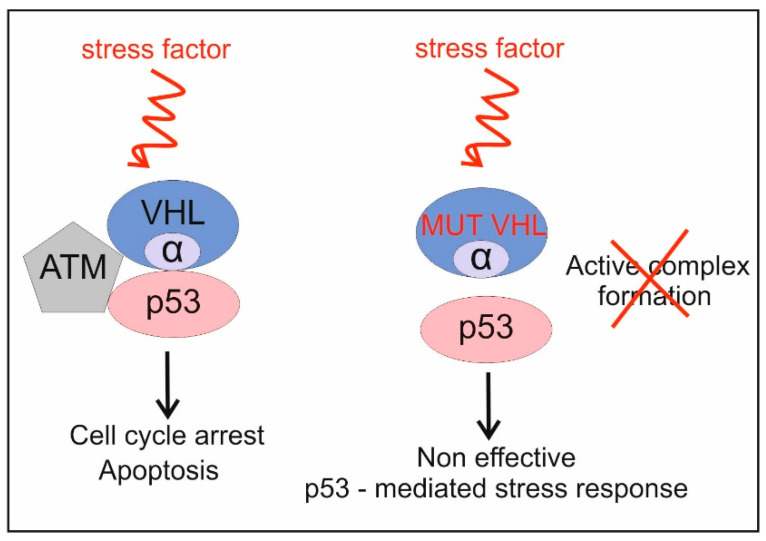
VHL interacts with p53, via its α domain, and also, with the ATM protein (Adapted from ref [21]). These interactions inhibit p53 degradation and eventually enhance the p53 response to stress. Mutations in the *VHL* gene probably result in the failure of the VHL–p53 interactions, and as a consequence, the p53-dependent stress response is inefficient.

**Figure 3 biomedicines-10-01330-f003:**
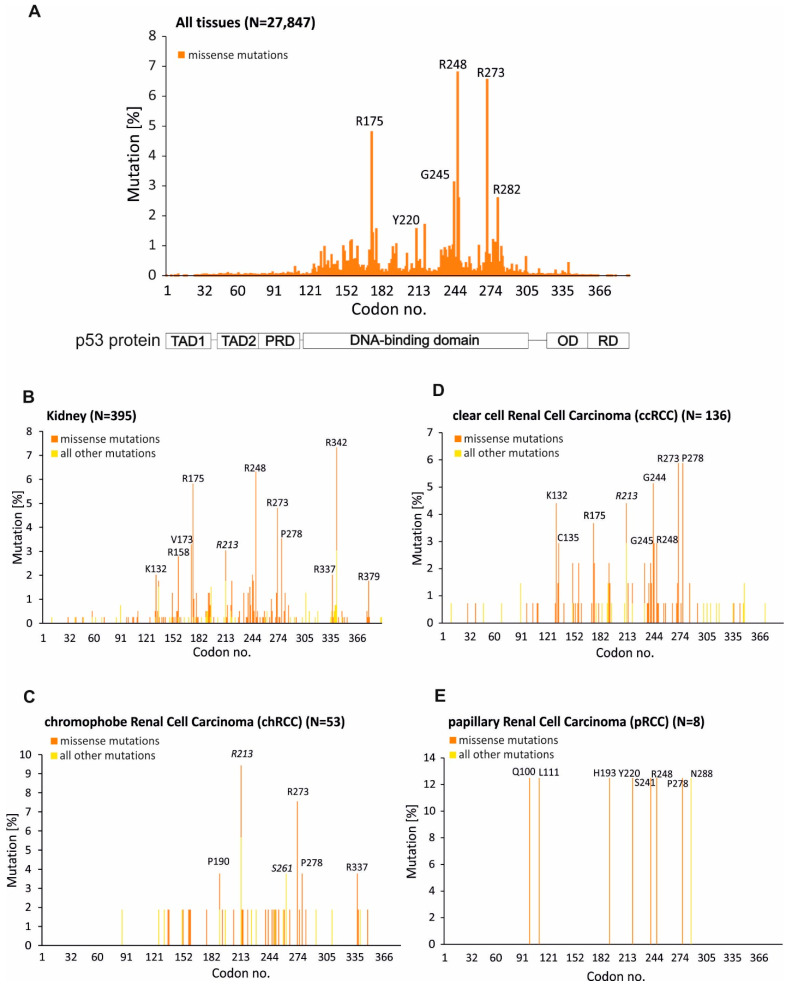
A graphical view of mutations across *Tp53* gene in all tissues (**A**), kidney (**B**), chromophobe renal cell carcinoma (**C**), clear cell renal cell carcinoma (**D**), and papillary renal cell carcinoma (**E**). The mutations are displayed at the amino acid level across the full-length of the gene. A schematic representation of the domains of p53 protein: TAD1 and TAD2—transactivation domains 1 and 2, PRD—proline rich domain, DNA-binding domain, OD—oligomerization domain, RD—regulatory domain (below **A**). Orange color indicates missense mutations whereas yellow color indicates all other mutations including nonsense mutations, frameshifts and deletions. N—total mutated samples taken for analysis (see Table 1). Data are derived from the *TP53* database (**A**) (https://tp53.isb-cgc.org/) and the COSMIC database (**B**–**E**).

**Figure 4 biomedicines-10-01330-f004:**
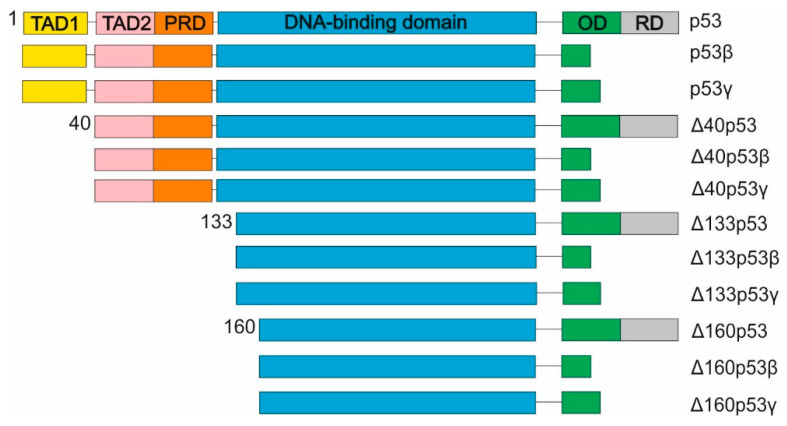
Human p53 isoforms. Twelve p53 isoforms are expressed from *Tp53* gene. There are four p53 isoforms differ from N-terminus (p53, Δ40p53, Δ133p53 and Δ160p53). Each of them possesses a distinct C-terminus resulting from different splicing events. The domains are described in Figure 3.

**Table 1 biomedicines-10-01330-t001:** Distribution of mutations in *Tp53*/p53 across kidney and sub-histological types of RCC based on data from the COSMIC database (https://cancer.sanger.ac.uk/cosmic/gene/, accessed on 1 May 2022).

	Kidney	ccRCC	chRCC	pRCC
Total samples tested	4766	2303	203	405
Total mutated samples	417 (395) *	155 (136) *	53	8
Total percentage of samples mutated [%]	8.75	6.73	25.62	1.98

*—Numbers in brackets indicate numbers of samples with mutations in the *Tp53* gene which are associated with defined mutations in p53 protein.

## Data Availability

Not applicable.

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
