# Peer review of "p53 and Its Isoforms in Renal Cell Carcinoma—Do They Matter?"

_biomedicines, 2022, doi:10.3390/biomedicines10061330_

Round 1

Reviewer 1 Report

Agata Swiatkowska present a quality and well-written review manuscript that is focused on P53 and its isoforms in renal cell carcinoma and questions if they matter.

Author tries to answer how do mutations in p53 influence the development and progression of renal cell carcinoma? The manuscript reviews the latest knowledge concerning the role of p53 and its mutants as well as potential functions of p53 isoforms in terms of development and progression of kidney cancer. The focus is on renal cell carcinoma as the most common kidney cancer.

Authors discusses wild-type and mutant p53 in renal cell carcinoma with respect to clinical prognosis for renal cell carcinoma patients.

Finally, author concludes that p53 and its isoforms seem to mediate cell response and cell sensitivity to treatment in renal cell carcinoma. However, more research is needed to better understand the role of network interactions of p53 and its isoforms with others proteins and RNAs in this cancer.

Overall, the manuscript is valuable for the scientific community and should be accepted for publication after edits are made.

===========================

Other comments:

1) Please check for typos throughout the manuscript.

2) The sole author of this manuscript uses “we” throughout the manuscript, i.e. “…we attempt to…”. Instead other linguistic constructs may be applied, i.e. “…attempt was made to…”.

3) Author is kindly encouraged to cite this article that overviews promising new tools for targeting p53 mutant cancers (with the focus on immunotherapy).

DOI: 10.3389/fimmu.2021.707734

Author Response

Dear Reviewer,

The response to the specific points made is set out below

1) Please check for typos throughout the manuscript.

2) The sole author of this manuscript uses “we” throughout the manuscript, i.e. “…we attempt to…”. Instead other linguistic constructs may be applied, i.e. “…attempt was made to…”.

I thank the Reviewer for comments. Following the Reviewer’s suggestion the manuscript was checked for typos and it was corrected. The linguistic constructs were changed as the Reviewer mentioned.

3) Author is kindly encouraged to cite this article that overviews promising new tools for targeting p53 mutant cancers (with the focus on immunotherapy).

DOI: 10.3389/fimmu.2021.707734

The proposed article is cited in the Conclusions section of the revised manuscript.

Kind Regards,

Agata Swiatkowska

Reviewer 2 Report

The review entitled " P53 and its isoforms in renal cell carcinoma. Do they matter?" written by Agata Swiatkowska describes the importance of p53 and its isoforms in the development and treatment of renal cell carcinoma. The review is nicely written and logically organized. I have no concerns and recommend the manuscript for publication after some minor suggestions have been addressed.

I felt that the manuscript would benefit from some more comprehensive discussions, particularly on p53 isoforms whose role in tumors is becoming more prominent. Adding more literature would help.

For example, I suggest to emphasize the role of Δ160p53 in cancer development and invasion (work by Tadijan et al, https://doi.org/10.3390/cancers13205231).  

The language should be improved, some sentences lack comprehension, also there are some grammatical incorrections.

Some abbreviations do not have explanations, eg. ADM, line 118.

Line 51. “ASR (age-standardized rate). The ASR…” unnecessary repetition.

P53 protein should be always be written “p53”.

The p53 database should have link, it is no more maintained by IARC.

I suggest to add the schematic figure of crosstalk between p53 and VHL.

Author Response

Dear Reviewer,

The response to the specific points made is set out below

1) I felt that the manuscript would benefit from some more comprehensive discussions, particularly on p53 isoforms whose role in tumors is becoming more prominent. Adding more literature would help. For example, I suggest to emphasize the role of Δ160p53 in cancer development and invasion (work by Tadijan et al, https://doi.org/10.3390/cancers13205231).  

I thank the Reviewer for suggestions. Data concerning Δ160p53 was added in the P53 isoforms in RCC section. More comprehensive discussion, which emphasizes the role of p53 isoforms as important factors in RCC therapy, is presented in the Conclusions section of the revised manuscript.

2) The language should be improved, some sentences lack comprehension, also there are some grammatical incorrections. Some abbreviations do not have explanations, eg. ADM, line 118. Line 51. “ASR (age-standardized rate). The ASR…” unnecessary repetition. P53 protein should be always be written “p53”. The p53 database should have link, it is no more maintained by IARC.

I thank the Reviewer for pointing out these mistakes. The abbreviations were explained. The grammar and typos were checked and corrected. The link to current p53 website is added in Figure 3 of the revised manuscript.

3) I suggest to add the schematic figure of crosstalk between p53 and VHL.

Following the Reviewer’s suggestion the schematic diagram showing the interaction of p53 and VHL is presented in Figure 2 of the revised manuscript.

Kind Regards,

Agata Swiatkowska

Reviewer 3 Report

This review article summarized the p53 status and role of its isoforms in RCC. Overall is clearly but still has several points needed to improve.

1.    The line 30-32 : “In 2020, there were 271, 249 new diagnosed cases with 115, 600 new death cases. Asia and Europe are two regions, with the highest rate of incidence, 156, 470 and 138, 611 reported cases, respectively, in 2020.” No reference showed in the above sentences. 

2.    Could Fig 2A also show the all other mutations as like as 2B-E?

3.    The line 347-351:“association of a high level of p53β with better prognosis have been observed for breast cancer and acute myeloid leukemia patients [49, 65]. Moreover, it has also been observed that p53β isoform plays an protective role in the case of breast cancer patients bearing mutation in p53 [49].” Please put the above sentences to line 287 which will become more better. 

4.    The line 353-353:”They have only observed that a smaller tumor size was characterized by Δ133p53 expression [25].” According reference 25, only 
p53â–³133 α (p=0.01) but hot p53â–³133β has (p=0.28) has real smaller tumor size comparing with yes or no expression. So line 353-353 should be changed to “They have only observed that a smaller tumor size was characterized byΔ133p53α expression [25].”

5.    The line 363-372 just repeated mention again of previous part. It better to discuss more things (ex: comparing different type of RCC with p53 different mutant patterns should have different type of treatment….)

Author Response

Dear Reviewer,

The response to the specific points made is set out below

1) The line 30-32 : “In 2020, there were 271, 249 new diagnosed cases with 115, 600 new death cases. Asia and Europe are two regions, with the highest rate of incidence, 156, 470 and 138, 611 reported cases, respectively, in 2020.” No reference showed in the above sentences. 

Thank the Reviewer for pointing out that no reference was shown. The link to website is added in the revised manuscript. 

2) Could Fig 2A also show the all other mutations as like as 2B-E?

The vast majority of mutations in Tp53 gene across all tissues are substitutions, particularly missense mutations (Baugh, Cell Death and Differentiation, 2018, 25, 154-160). My idea was to compare the pattern of p53 missense mutations with the p53 mutations’ profiles of sub-types of RCC. Such comparison allows to demonstrate how unique or common p53 mutations are in each of RCC sub-types. For this reason I would like to keep the graph unchanged.

3) The line 347-351:“association of a high level of p53β with better prognosis have been observed for breast cancer and acute myeloid leukemia patients [49, 65]. Moreover, it has also been observed that p53β isoform plays an protective role in the case of breast cancer patients bearing mutation in p53 [49].” Please put the above sentences to line 287 which will become more better. 

Following the Reviewer suggestion the pointed fragment of text is rearranged in the revised manuscript.

4) The line 353-353:”They have only observed that a smaller tumor size was characterized by Δ133p53 expression [25].” According reference 25, only 
p53
â–³133 α (p=0.01) but hot p53â–³133β has (p=0.28) has real smaller tumor size comparing with yes or no expression. So line 353-353 should be changed to They have only observed that a smaller tumor size was characterized byΔ133p53α expression [25].”

The Reviewer is correct. However, both names Δ133p53 and Δ133p53α can be used interchangeably because they refer to p53 isoform which is translated from the methionine codon 133 and produced through splicing from exon 9 to exon 10 (Marcel, Oncogene, 29, 2691–2700, 2010). Since discussed p53 isoform is called Δ133p53α in a cited article, as the Reviewer pointed, this name of isoform is changed in the revised manuscript.

5) The line 363-372 just repeated mention again of previous part. It better to discuss more things (ex: comparing different type of RCC with p53 different mutant patterns should have different type of treatment….)

Thank the Reviewer for comments. Following the Reviewer’s suggestion potential new RCC therapies based on antisense oligomers, small molecules or targeting p53 mutants are discussed in the Conclusion section in the revised manuscript.

Kind Regards,

Agata Swiatkowska